# The Prevalence of Lower Limb and Genital Lymphedema after Prostate Cancer Treatment: A Systematic Review

**DOI:** 10.3390/cancers14225667

**Published:** 2022-11-18

**Authors:** Andries Clinckaert, Klaas Callens, Anne Cooreman, Annabel Bijnens, Lisa Moris, Charlotte Van Calster, Inge Geraerts, Steven Joniau, Wouter Everaerts

**Affiliations:** 1Department of Cellular and Molecular Medicine, University of Leuven, 3000 Leuven, Belgium; 2Department of Urology, University Hospitals of Leuven, 3000 Leuven, Belgium; 3Department of Rehabilitation Sciences, University of Leuven, 3000 Leuven, Belgium; 4Department of Development and Regeneration, University of Leuven, 3000 Leuven, Belgium

**Keywords:** lower limb lymphedema, genital lymphedema, prostate cancer, radical prostatectomy, pelvic lymph node dissection, external beam radiotherapy

## Abstract

**Simple Summary:**

Prostate cancer patients that receive treatment (surgery of radiation therapy) directed to the pelvic lymph nodes may suffer from secondary lymphedema in the lower limbs and/or the genital area. Despite its potential impact on quality of life, reports on secondary lymphedema after prostate cancer therapy are scarce and prevalence rates vary between different studies. Here we perform a systematic literature search to estimate the prevalence of lymphedema after surgery, radiation therapy, or both, to the pelvic lymph nodes in men with prostate cancer.

**Abstract:**

(1) Background: Secondary lymphedema is a chronic, progressive, and debilitating condition with an important impact on quality of life. Lymphedema is a frequently reported complication in oncological surgery but has not been systematically studied in the setting of prostate cancer. (2) Methods: Pubmed/MEDLINE and Embase were systematically searched to identify articles reporting on lower limb or genital lymphedema after primary treatment (surgery of radiation therapy) of the prostate and the pelvic lymph nodes in men with prostate cancer. Primary outcome was the prevalence of lower limb and genital lymphedema. (3) Results: Eighteen articles were eligible for qualitative synthesis. Risk of bias was high in all included studies, with only one study providing a prespecified definition of secondary lymphedema. Eleven studies report the prevalence of lower limb (0–14%) and genital (0–1%) lymphedema after radical prostatectomy with pelvic lymph node dissection (PLND) Seven studies report a low prevalence of lower limb (0–9%) and genital (0–8%) lymphedema after irradiation of the pelvic lymph nodes. However, in the patient subgroups that underwent pelvic irradiation after staging pelvic lymph node dissections, the prevalence of lower limb (18–29%) and genital (2–22%) lymphedema is substantially elevated. (4) Conclusion: Prostate cancer patients undergoing surgery or irradiation of the pelvic lymph nodes are at risk of developing secondary lymphedema in the lower limbs and the genital region. Patients receiving pelvic radiation after pelvic lymph node dissection have the highest prevalence of lymphedema. The lack of a uniform definition and standardized diagnostic criteria for lower limb and genital lymphedema hampers the accurate estimation of their true prevalence. Future clinicals trials are needed to specifically evaluate secondary lymphedema in patients undergoing prostate cancer treatments, to identify potential risk factors and to determine the impact on quality of life.

## 1. Introduction

Secondary lymphedema is a well-known complication of cancer therapy. In men undergoing prostate cancer treatment, surgical resection or irradiation of the pelvic lymph nodes can result in lymphedema (LE) of the lower limbs and the scrotal and suprapubic regions.

Lymphedema results from damage to the lymphatic system causing accumulation of fluid and plasma proteins in the interstitial compartment, adipose deposition, chronic tissue inflammation and fibrosis [1,2,3]. Clinical symptoms include abnormal tissue swelling, sensation of limb heaviness, erythema, pain, and impaired limb function [2,4],resulting in a negative impact on quality of life (QoL) [5]. When diagnosed at an early stage, lymphedema can be treated with physical therapy and compression. However, when left untreated, lymphedema can deteriorate over time and become more difficult to treat.

Therefore, a better understanding of the prevalence of secondary LE after prostate cancer therapy is important for pre-operative counseling of patients and identifying the needs for post-operative lymphedema therapies. Several studies have evaluated the prevalence (between 0–50%) of secondary LE and potential risk factors for LE after therapies for breast and other gynecological cancers [1,6,7,8]. In contrast, secondary LE in the setting of prostate cancer has not been systematically studied.

This study aims to systematically review the literature, reporting on the prevalence of lower limb and genital LE in patients undergoing surgical resection or irradiation of the pelvic lymph nodes in patients with prostate cancer.

## 2. Materials and Methods

### 2.1. Search Strategy and Evidence Acquisition

A systematic review of the medical literature following the Preferred Reporting Items for Systematic Reviews and Meta-Analysis (PRISMA) guidelines was conducted in July 2019 and updated in August 2022 [9]. The detailed study protocol for this review has been registered online with PROSPERO (CRD42020163864). Databases including MEDLINE/Pubmed and Embase were systematically searched for English articles reporting LE after PCa treatment. The following index terms (including synonyms) were used: “prostate cancer “prostatectomy “lymph node dissection”, “radiotherapy”, “lymphatic irradiation”, “lymphedema”, “complication”, “postoperative edema”. The term “complication” was included in our search strategy to avoid missing articles that report “lymphedema” only in the full text results, but not in the abstract or key words.

Articles were eligible for inclusion if (1) the article was published between 1 January 1980 and August 2022, (2) at least 50 patients were included, (3) participants were male adults (aged 18 years or more) with histologically proven PCa, (4) patients received any of the following primary intervention: radical prostatectomy (RP) (all routes and approaches) with standard or extended Pelvic Lymph Node Dissection (PLND), or Radiation Therapy (RT) of prostate and pelvis irrespective of (neo)adjuvant androgen deprivation therapy, (5) outcomes on lower limb or genital LE were reported. Control groups were patients receiving RP with limited or no PLND, or patients receiving prostate-only irradiation. Non-English language articles, case reports and reviews were excluded.

Relevant systematic reviews were inspected for potentially relevant studies but were not included for qualitative synthesis. The absence of a comparator group was no exclusion criterion. We excluded articles published before 1980, since it was around this time that Walsh described the “modern” radical retropubic prostatectomy technique [10].

After removal of duplicates, abstracts and retrieved full texts were independently screened for eligibility in duplicate by two authors (KC, AC). Any disagreements or uncertainties were resolved by discussion or reference to an independent third party (LM). After full text screening, data extraction was performed in duplicate by the same two reviewers (KC, AC). Disagreements were this time discussed in consensus, and when necessary, a third party (LM) was consulted.

Data were extracted according to a predefined data extraction template, consisting of study details, patient characteristics (sample size, follow-up, age, initial Prostate Specific Antigen (iPSA), biopsy Gleason Score (bGS), clinical TNM stage, pathological Gleason Score (pGS), pathological TNM stage, number of lymph nodes dissected, number of positive lymph nodes, tumor risk category, race, comorbidities, Body Mass Index (BMI) and prostate volume), intervention characteristics (surgery/RT, route of surgery, PLND performed & template used, type and dose of RT, neo-adjuvant or adjuvant treatment, chemotherapy) and outcomes (development of LE, QoL).

### 2.2. Outcome Measurement

The primary outcome measurement is the prevalence of lower limb, genital or suprapubic LE. The definitions of LE provided by the authors were used but LE needed to be reported as a separate entity. A secondary outcome is to evaluate potential risk factors for secondary lymphedema (if described).

### 2.3. Risk of Bias and Study Quality Assessment

To assess the validity of the included studies we used The Cochrane Handbook for Systematic Reviews [11]. We judged the risk of bias (RoB) from each included study as ‘high’, ‘low’, or ‘unclear’ for the following seven individual items: random sequence generation (selection bias), allocation concealment (selection bias), blinding of participants (performance bias), blinding of outcome assessor (detection bias), completeness of outcome data reporting (attrition bias), selective reporting (reporting bias), and other possible sources of bias such as conflicts of interest.

## 3. Results

### 3.1. Study Selection

Our initial electronic database search identified 10,561 records (Figure 1). After removing duplicates and screening all titles and abstracts, 142 trials were scrutinized for further eligibility. Of those, eighteen articles met our eligibility criteria and were consequently included in our qualitative analysis. Most records were excluded because they did not report lower limb or genital LE as a separate outcome.

### 3.2. Study Characteristics

Table 1 shows baseline study characteristics from each included study, organized by primary intervention (radical prostatectomy versus external beam radiotherapy (EBRT). All studies were published between 1980 and 2022. Of the nineteen studies (in eighteen papers) included, three were randomized controlled trials (RCT), 4 were prospective comparative studies, 4 were prospective observational studies, and 8 were retrospective non-randomized trials. Sample size ranged from 99 to 3675 with a total of 9223 participants included in this qualitative analysis. Median age ranged from 61 to 68 years of age.

### 3.3. Risk of Bias within Studies

Figure 2 and Figure 3 outline the Risk of Bias (RoB) assessment of all the included studies. Overall, the RoB within included studies was considered very high. Since only three RCTs were included, there was a high risk of selection, detection and performance bias. Most studies had a low or moderate RoB regarding attrition bias. Reporting bias was rated as high, with only one study that predefined lymphedema in its methods [13]. Other sources of bias were often unclear.

### 3.4. Lower Limb Lymphedema

All the included studies report the prevalence of lower limb LE, with a prevalence ranging from 0% and 14% (Table 2). Importantly, only the LAPPRO study provides a prespecified definition of lower limb LE and the methodology for assessment of LE [13]. In this study, the authors use a standardized questionnaire with two specific questions to determine patient-reported “swelling in the left/right groin” and “swelling in the left/right leg” at three months after surgery. In addition, they also describe staff-reported LE at different time points after surgery. It is unclear how lower limb LE is determined in the other included studies. The bubble graph in Figure 4 depicts the prevalence of lower limb lymphedema in the included studies from 1980 to 2022.

#### 3.4.1. Surgery

The prevalence of secondary lower limb LE after pelvic lymph node dissection ranged from 0 to 14% (Table 2). Five studies compare LE after RP with extended PLND versus RP with limited PLND [13,15,16,20,22]. Only Morizane et al., found a statistically significant difference in the rate of LE with 6% (28/431 patients) LE in patients undergoing extended PLND versus 1% (7/902 patients) in the limited PLND group (*p* < 0.001) [20]. Four studies without comparator group evaluate the prevalence of lower limb LE after RP with extended PLND [17,18,19,21]. In these studies, lymphedema is observed in 2–10% of patients. The highest prevalence of lower limb LE is reported in the LAPPRO trial, which reports patient-reported outcomes. Importantly, patient-reported prevalence (14%, 85/621 patients) of lower limb LE in this study is considerably higher than staff-reported LE rates (5%, 32/616 patients).

#### 3.4.2. External Beam Radiotherapy with or without Staging PLND

Seven manuscripts (reporting on eight trials) report the prevalence of lower limb LE after RT to the prostate and the pelvic lymph node regions, with lymphedema rates ranging from 0% to 9% (Table 2) [23,24,25,26,27,28,29]. Four studies specifically report the prevalence of LE in subgroups that underwent staging PLND followed by irradiation of the pelvic lymph nodes in case of pathological lymph node involvement [26,27,28,29]. In these subgroups, the prevalence of secondary lymphedema (18–29%) is considerably higher than in subgroups that did not undergo staging PLND (0–8%).

### 3.5. Genital Lymphedema

Only a few studies make a distinction between lower limb and genital LE (Table 2). A description of the methodology to assess genital LE is lacking in all included studies. Genital LE as a separate entity is reported in 0% to 22% of patients [21,22,26,27,28,29].

#### 3.5.1. Surgery

Porcaro et al., reports only one out of 211 patients (0.5%) suffering from scrotal edema after RP with ePLND [21]. In a prospective observational study, Yuh et al., describe scrotal edema in 1.5% (3/204) of patients undergoing RP with extended PLND, and 0.5% (1/202) of patients undergoing RP with limited PLND [22].

#### 3.5.2. External Beam Radiotherapy with or without Staging PLND

Five radiotherapy studies report the prevalence of genital LE (Table 2) [24,26,27,28,29]. Aristizabal et al., report scrotal or penile LE in 2% (4/218) of patients treated with external beam radiotherapy only [24]. In Perez et al., genital LE is observed in 4 of 195 patients (2%) of which 14 patients received a staging laparotomy [27]. Scrotal and penile LE was observed by Pilepich et al., in 6 of 267 patients (2%), all of which underwent a staging PLND before whole pelvis irradiation [28]. In the RTOG75-06 and RTOG-77 trials, genital LE is reported in 0 to 6% of patients; with higher lymphedema rates in the subgroup that underwent staging PLND [29]. The highest prevalence of genital LE is reported by Forman et al., in 22% (9/41) of patients that underwent pelvic EBRT following a staging PLND versus only 1% (2/199) in patients who did not undergo staging PLND [26].

## 4. Discussion

Secondary lymphedema can be a major concern for patients undergoing oncological therapy, causing discomfort, functional impairment, and even psychosocial distress [4]. Most data from quality of life and medical costs are derived from upper limb lymphedema in women undergoing breast cancer treatment [4], whereas data from prostate cancer patients are sparse. Here, we performed a systematic literature review to determine the prevalence of secondary lymphedema in prostate cancer patients undergoing primary treatment of the prostate and the pelvic lymph nodes with surgery and/or radiation therapy.

In this systematic review, we found the rate of secondary LE ranging from zero to fourteen percent in patients undergoing PLND and from zero to eight percent in patients undergoing pelvic nodal irradiation. Importantly the prevalence of secondary LE is much higher in the subgroups that underwent pelvic nodal irradiation after staging PLND (between 18 and 29%) suggesting that the cumulative effect of surgery and irradiation results in substantially higher LE rates. PLND is considered the most sensitive technique to determine microscopic lymph node involvement, but the oncological benefits of this procedure remain elusive [30,31]. Since performing a PLND is not only associated with potential peri- and postoperative complications, including lymphoceles, thromboembolic events and neurovascular injuries [32], but also with the long-term risk of lower limb and genital edema, careful preoperative patient selection and counseling are crucial.

In this review, the reported LE prevalence varies considerably between different studies. These differences depend on differences in patient selection, differences in technique (e.g., extend of PLND) as well as differences in lymphedema assessment between different studies. The International Society of Lymphology defines Lymphedema as the ‘external manifestation of lymphatic system insufficiency and deranged lymph transport.’ The detection of lymphedema can be clinically evident in patients with clinically measurable swelling but can be more tedious in patients with subjective perceptions of swelling and/or limb heaviness without a clinically detectable swelling. Therefore, the diagnosis of lymphedema depends on patient-reported symptoms, visual inspection, skin palpation and measurements of volume differences between both limbs [1,33,34,35].

The LAPPRO trial [13] was the only included study that performed a standardized assessment of postoperative LE. Lymphedema was defined as patient-reported “swelling in the left/right groin” and “swelling in the left/right leg” using a standardized questionnaire at 3 months after surgery. The authors also recorded staff-reported lymphedema, but no objective measurements were performed. Interestingly, the rate of patient-reported swelling (14%) at 3 months was almost threefold higher than staff-reported swelling (4%), suggesting an underreporting on staff reports. In all other studies a clear definition of LE or the methodology of how LE was determined is completely lacking. Therefore, the reported rates of secondary lymphedema might represent an underestimation of the true prevalence.

In the context of breast cancer treatments, LE is a well-known complication [4]. Several risk factors have been identified, including axillary lymph-node dissection, adjuvant RT, and high BMI, and several risk models have been developed to predict upper limb LE [35,36]. Moreover, there is a remarkable awareness for health-related QoL in these patients with routine use of patient-reported outcome measurements [33]. In contrast, no risk factors, other than performing a PLND have been identified as a risk factor for lower limb LE in PCa patients [13]. Although Morizane et al. [20] found a significantly higher prevalence of lower limb LE in patients undergoing extended versus limited PLND, Carlsson et al., did not find a correlation between the number of lymph nodes removed and the prevalence of secondary LE [13].

It is remarkable that, compared to breast cancer, secondary LE in prostate cancer patients has received little attention. A possible explanation could be the lower prevalence of lower limb LE in men undergoing prostate cancer treatments (0–14%) compared to upper limb LE in women undergoing breast cancer therapies (14–40%) [1]. Moreover, the functional and cosmetic aspects of LE may receive more attention in breast cancer, whereas sexual and urinary function are the main focus of attention in PCa patients [37]. Another reason could be the difficulty to objectivize lower limb LE when both limbs are affected. In patients with unilateral breast cancer, volume and circumference measurements of the affected limb, can easily be compared to the limb on the untreated side. In contract, PCa patients usually undergo bilateral PLND hereby affecting lymphatic transport in both limbs. Moreover, bilateral measurements can be biased by muscle hypertrophy or weight gain, equally affecting both limbs. The use of techniques that evaluate edema in a direct way, such a bio-impedance spectroscopy and tissue dielectric constant measurements, can assist in the diagnosis of LE, but these techniques have not been validated in the setting of lower limb or genital LE [38,39,40].

## 5. Limitations of This Study

Despite our systematic methodology, this review has several limitations. First, only a limited number of studies report on our outcomes of interest. Second, there is a lack of standardization in the definitions of LE and the methodology to determine the presence of lower limb and genital LE. Moreover, details about the time course of lymphedema are lacking in all but one study. As such, most included studies had a high RoB. Third, there is substantial heterogeneity between studies considering the proportion of patients undergoing staging PLND, surgical (open versus robot-assisted, extend of PLND) and radiation techniques (the template, duration, total dose). Moreover, outcomes of pelvic irradiation were published between 1980 and 1997, which may limit the translation to modern radiotherapy techniques [41].The lack of a unified definition of LE and the heterogeneity of the included studies withheld us from performing a meta-analysis.

## 6. Conclusions

This review systematically analyzes the published literature to determine the prevalence of lower limb and genital LE in PCa patients undergoing surgery or irradiation of the pelvic lymph nodes. The prevalence of lymphedema in the lower limbs and genital regions range from 0–14% and 0–1% after surgery, and 0–9% and 0–8% after pelvic radiation respectively, with a much higher prevalence in patients that underwent PLND followed by pelvic radiotherapy (18–29% and 2–22%). The great heterogeneity between different studies can be attributed to a lack of a standardized definition, a lack of standardized assessment tools and the absence of well-designed prospective studies to assess secondary lymphedema and its impact on quality of life. For PCa patients, LE is still the ‘forgotten vascular disease’ [42].

## Figures and Tables

**Figure 1 cancers-14-05667-f001:**
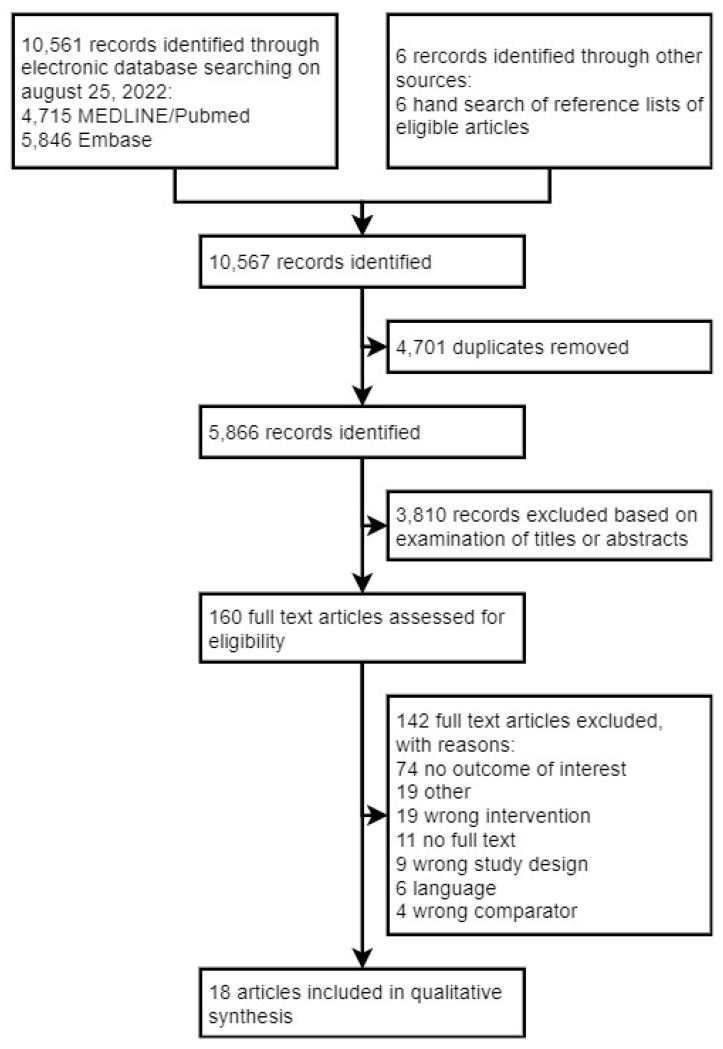
Study selection flow diagram according to the Preferred Reporting Items for Systematic reviews and Meta-analyses (PRISMA) guidelines.

**Figure 2 cancers-14-05667-f002:**
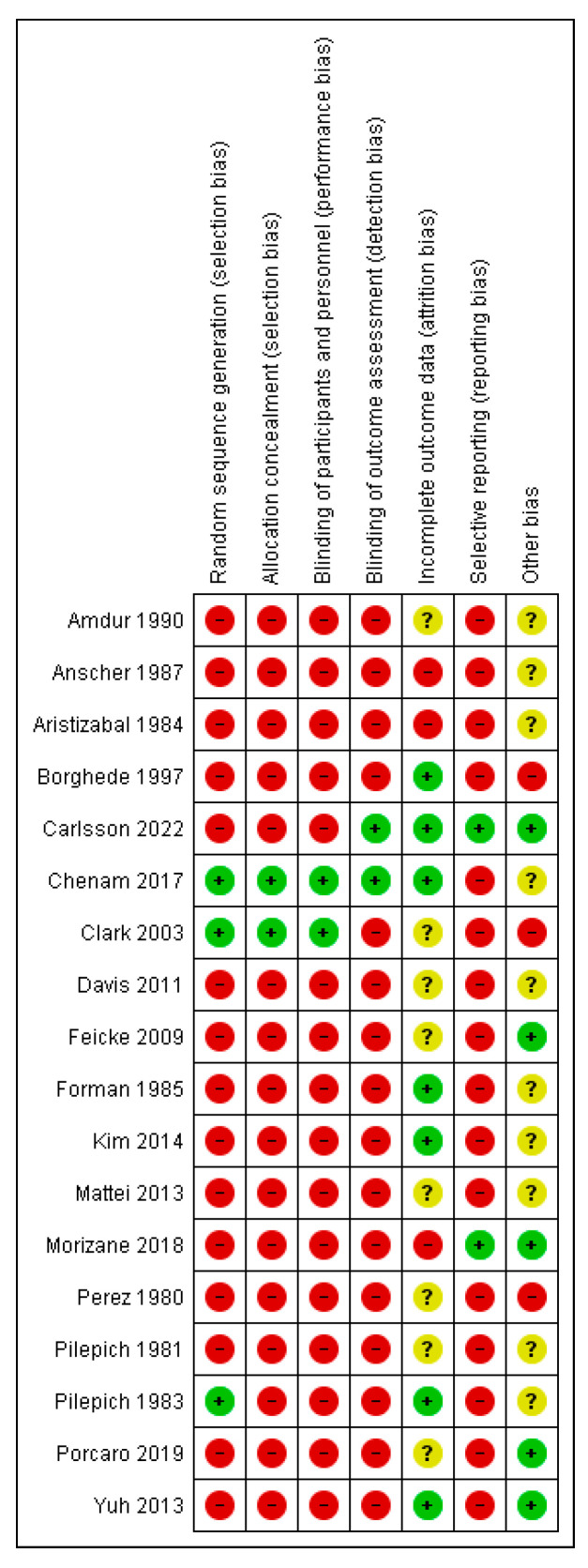
Risk of bias summary representing the author’s judgement about each risk of bias topic for each included study.

**Figure 3 cancers-14-05667-f003:**
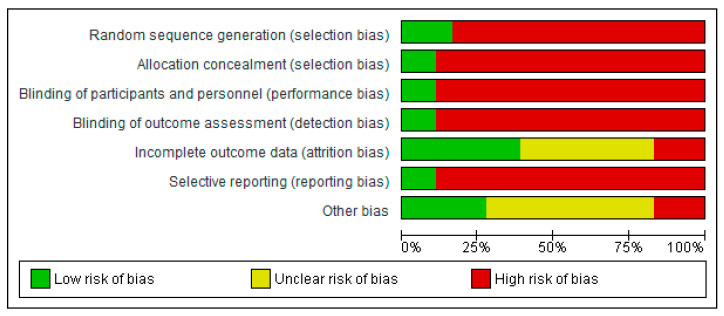
Risk of bias graph representing author’s judgement about each risk of bias item presented as percentage of risk across all studies.

**Figure 4 cancers-14-05667-f004:**
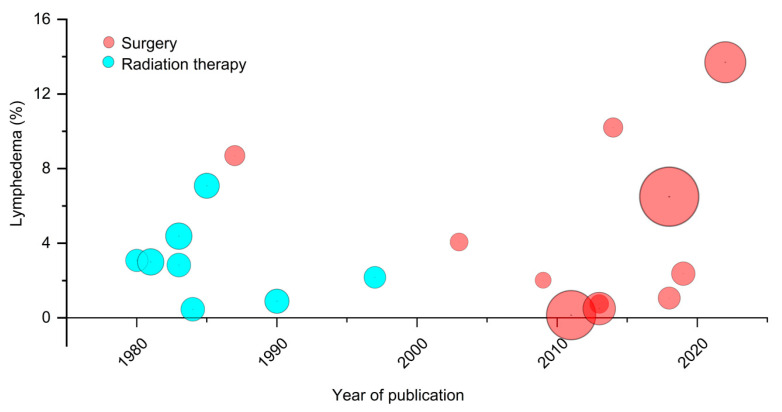
Bubble plots, depicting the prevalence of lower limb lymphedema for surgery (blue) and radiation therapy (orange) over time. Bubble area corresponds to the sample size.

**Table 1 cancers-14-05667-t001:** Overview of included studies.

Study ID; Country; Design; Recruitment Period	Treatment	Patients (N)	FU	Age (Years) (Mean/Median/IQR/Range)	iPSA (Mean/Median/IQR/Range)	bGS (N, %)	cT Stage (N, %)	RP Type (Robot/Laparoscopy/Open) PLND (Template, N, %)	Dose (Gy) Pelvic RT (N/%)	Neoadjuvant Therapy (Type/%) Adjuvant Therapy (Type/%)	LN Removed (Mean/Median/IQR/Range) pN1 (N, %)	Comorbidities
**Surgery**												
Anscher MS [12], 1987, USA, retrospective comparative 1970–1983	RP ± PLND	113	15 years	64 (range: 40–78)	NR	Histological differentiation grade: Well: 16 (14%), Moderate: 62 (55%), Poor: 23 (20%), NR: 12 (11%)	Whitmore stage: A: 20 (18%) B: 84 (74%) C: 8 (7%) D: 1 (1%)	retropubic: 25 (22%), perineal: 88 (78%) PLND: 77 (68%)	NA	ADT: 69 (62%)	pN1: 3/77 patients (4%)	NR
	RP ± PLND + EBRT	46	15 years	61 (range: 43–77)	NR	Histological differentiation grade: Well: 7 (15%), Moderate: 27 (59%), Poor: 9 (20%), NR: 3 (6%)	Whitmore stage: A: 8 (17%) B: 35 (76%) C: 2 (4%) D: 1 (3%)	retropubic 9 (20%), perineal 37 (80%) PLND: 39 (85%)	45 to 50 Gy to the whole pelvis + 10 to 15 Gy boost on prostatic bed	ADT: 8 (17%)	pN1: 4/39 (10%)	
Carlsson S [13], 2022, prospective non-randomized controlled trial 2008–2011	RP ± PLND vs RARP ± PLND	3675	3 months	NR	NR	NR	NR	PLND 645 (18%)	NA	NA	NR	*NR*
Chenam A [14], 2018, USA, RCT 2012–2016	RARP ± limited or extended PLND + no pelvic drain	92	90 days	634 (IQR: 57–69)	6.2 (IQR: 4.7–7.8)	≤6: 27 (29%) 7: 50 (54%) ≥8: 15 (16%)	cT1: 54 (59%) cT2: 35 (38%) cT3: 3 (3%)	Robot PLND: None: 11 (12%) Limited: 16 (17%) Extended: 65 (71%)	NA	NR	17 pN1: 6 (7%)	*BMI*: *28.6 (IQR: 26.0–30.8)*
	RARP ± limited or extended PLND + pelvic drain	97	90 days	65 (IQR: 58–69)	5.8 (IQR: 4.5–8.4)	≤6: 19 (20%) 7: 65 (67%) ≥8: 13 (13%)	cT1: 58 (60%) cT2: 34 (35%) cT3: 5 (5%)	Robot PNLD: None: 9 (9%) Limited: 11 (11%) Extended: 77 (79%)	NA	NR	18 pN1: 13 (13%)	*BMI*: *28.7 (IQR: 25.9–31.1)*
Clark T [15], 2003, USA, RCT NR	RRP + limited PLND (ipsilateral)	123*	NR	61 (range: 45–75)	Mean: 7.4 ng/ml	≤6:83 (68%), 7: 25 (20%), ≥8: 15 (12%)	cT1c: 88 (72%) cT2a: 26 (21%) cT2b: 7 (5.7%) cT3: 2 (1.3%)	Open, retropubic PLND: limited	NA	NR	pN1: 3 (2%)	NR
	RRP + ePLND (contralateral)	123*	NR	61 (range: 45–75)	Mean 7.4 ng/ml	≤6: 83 (68%), 7: 25 (20%) ≥8: 15 (12%)	cT1c: 88 (72%) cT2a: 26 (21%) cT2b: 7 (5.7%) cT3 (1%)	open, retropubic PLND: extended	NA	NR	pN1: 4 (3%)	NR
Davis JW [16], 2011, USA, prospective comparative 2006–2010	RARP + limited PLND	261	18 months	NR	NR	NR	NR	Robot PLND: limited	NA	NR	8 (IQR: 5–11) pN1: 7%	NR
	RARP + ePLND	670	36 months	NR	NR	NR	NR	RARP PLND: extended	NA	NR	16 (IQR: 11–21) pN1: 18%	NR
Feicke A [17], 2008, Switzerland, retrospective descriptive 2006–2008	RARP + ePLND	99	NR	64 (range: 45–78)	7.7 (range: 1.5–84.6)	5: 2 (2%), 6: 18 (18%), 7: 64 (65%), 8: 8 (8%), 9: 5 (5%), NR: 2 (2%)	cT1: 66 (67%) cT2: 27 (27%) cT3: 6 (6%)	Robot PLND: extended	NA	Neo-adjuvant ADT: 2 patients	19 (range: 8–53) pN1: 16 (16%)	*BMI*: 26.4 (range: 19.8–34.3)
Kim KH [18], 2014, Korea, prospective observational 2008–2011	RARP + ePLND	147	NR	66 (IQR: 62–70)	10.7 (IQR: 6.5–17.4)	6: 19 (12.9%), 7: 57 (38.8%), 8–10: 71 (48.3%)	cT1: 80 (54.4%) cT2: 44 (29.9%) cT3: 23 (15.7%)	Robot PLND: extended	NA	NR	22 (18–26) pN1: 24 (16%)	*BMI*: 24.2 (IQR: 22.4–25.6)
Mattei A [19], 2013, Switzerland & Italy, prospective observational 2008–2011	RARP + ePLND	134	3 months	64 (IQR: 59–68)	8.6 (IQR: 6.1–13.5)	6: 33 (24.6%), 7: 76 (56.8%), 8–10: 25 (18.6%)	cT1c: 60 (44.8%) cT2a-T2b: 72 (53.7%) cT3: 2 (1.5%)	Robot PLND: extended	NA	NR	14 (11–19) pN1: 18 (13%)	NR
Morizane S [20], 2018, Japan, retrospective comparative 2010–2015	RARP + limited PLND	902	28 days	66 (IQR: 62–71)	7.8 (IQR: 5.6–11.4)	6: 147 (16.3%), 7: 536 (59.4%), 8: 110 (12.2%), ≥ 9: 109 (12.1%)	cT1: 381 (42.2%) cT2: 454 (34.1%) cT3: 61 (6.8%)	Robot PLND: limited	NA	NR	5.0 (3.0–8.0) pN1: 5 (1%)	*BMI*: 23.6 (22.0–25.4)
	RARP + ePLND	431	28 days	67.0 (IQR: 63.0–71.0)	7.3 (IQR: 5.4–10.4)	6: 5 (1.2%), 7: 123 (28.5%), 8: 159 (36.9%), ≥ 9: 144 (33.4%)	cT1: 48 (11.1%) cT2: 279 (64.7%) cT3: 98 (22.7%)	Robot PLND: extended	NA	NR	19.0 (14.0–24.0) pN1: 53 (12%)	*BMI*: 23.3 (21.8–25.3)
Porcaro AB [21], 2019, Italy, retrospective descriptive 2013–2017	RARP + ePLND	211	4 months	65 (IQR: 61–70)	7 (IQR: 4.9–9.9)	>7: 44 (20.9%)	cT1: 142 cT2–3: 69	Robot PLND: extended	NA	NR	26 (21–33) pN1: 28 (13%):	*BMI*: *25.3 (23.5–28.0)*
Yuh BE [22], 2013, USA, prospective comparative 2008–2012	RARP + limited PLND	204	90 days	64 (IQR: 58–70)	5.9 (IQR: 4.4–9.1)	6: 13 (6.4%), 3 + 4: 112 (54.9%), 4 + 3: 45 (22.1%), 8: 25 (12.2%), 9: 9 (4.4%)	cT1: 147 (72.1%), cT2: 56 (27.4%), cT3: 1 (0.5%)	Robot PLND: limited	NA	NR	7 (5–9) pN1: 8 (4%)	*BMI*: 27.5 (IQR: 25.2–30.3)
	RARP + ePLND	202	90 days	64 (IQR: 58–69)	5.5 (IQR: 4.2–8.3)	6: 12 (5.9%), 3 + 4: 121 (59.9%), 4 + 3: 40 (19.8%), 8: 23 (11.4%), 9: 6 (3.0%)	cT1: 139 (68.8%), cT2: 61 (30.2%) cT3: 2 (1.0%)	Robot PLND: extended	NA	NR	21.5 (17–27) pN1 24 (12%)	*BMI*: 27.1 (IQR: 25.2–30.5)
Amdur RJ [23], 1990, USA, retrospective descriptive 1964–1982	EBRT ± pelvic RT	225	> 5 years	66 (range: 45–81)	NR	*Whitmore stage, histological grade*: Well: 84 (37%) Moderate 97 (43%) Poor 37 (16%) N.R. 7 (3%)	*Whitmore stage*: A: 27 (12%) B: 87 (39%) C: 111 (49%)	EBRT PLND: Limited 16 (7%)	Stage A- B1: 6500 cGy in 7–7.5 weeks Stage B2-C: 6500–7000 cGy in 7–8.5 weeks. Pelvic RT: 214 (95%).	No	NR	NR
**Radiation Therapy**												
Aristizabal SA [24], 1984, USA, retrospective descriptive 1972–1979	EBRT prostate ± pelvic RT	218	>36 months	68 (range: 48–89)	NR	NR	NR A2: 17 (1%) B: 101 (5%) C: 82 (10%) D1: 18 (3%)	PLND: Limited 9 (4%)	6500–7000 rad in 6–7 weeks (n = 184) 4600–5000 (n = 3) 300 rad 3×/week for 6–7 weeks (n = 31). WPRT: 58 pts (32%)	5 underwent RP first	NR	
Borghede G [25], 1997, Sweden, prospective observational 1987–1992	EBRT prostate ± pelvis	184	46 months (24–96)	67 (range: 46–83)	NR	*WHO classification*: well: 37 (20%) moderate: 84 (46%) poor: 63 (11%)	*AUAC clinical staging:* A1: 1 (1%) A2: 10 (5%) B1: 62 (34%) B2: 14 (8%) C1: 65 (35%) C2: 32. (17%)	PLND: Limited: 154 (84%),	Dose: First 161 patients: 70 Gy, 2.0 Gy 5×/week in 7 weeks; last 23 patients: 64.8 Gy; 2.4 Gy 4×/week in 7 wks. WPRT: 161 (88%)	NR	range 1–12.	
Forman [26] 1985, USA, Prospective observational 1975–1983	EBRT prostate + pelvis	240	median 40 months (range 1–9 years)	68 (52–86)	NR	2–4: 23 (11%) 5: 33 (16%); 6: 60 (29%), 7: 45 (22%); 8: 25 (12%), 9–10: 18 (9%), NR: 36 (15%)	Whitmore staging system: A2: 27 (13%), B1: 29 (14%), B2: 45 (22%), C: 103 (51%)	PLND: Limited 41 (17%)	Total dose to the prostate tumor 6500 rad.	16 radical suprapubic prostatectomies before EBRT	NA	NR
Perez [27] 1980, USA, Retrospective descriptive 1966–1975	EBRT prostate + pelvis	195	mean 4.6 y	NR	NA	degree of differentiation: Well 75 (38%), Moderate 72 (26%), Poor 41(21%) NR 6 (3%).	Whitmore staging: B: 42 (22%) C: 141(72%) D1: 12 (6%)	PLND 14 (7%)	5000 rad to midplane pelvis. 6000 to 7000 rad to prostate. dose fractionation: 180 rad/day, 5×/week. Para-Aortic radiation 6 (3%)	ADT 25 (13%)	NA	NR
Pilepich [28] 1981, USA, Retrospective descriptive 1967–1978	EBRT Prostate + pelvis	267	median 48 months (mean 58 months)	NA	NA	NA	Whitmore staging A: 6 (2%) B: 72 (27%) C: 173 (65%) D: 16 (6%)	PLND: 31 (12%)	whole pelvis: 5000 rad in 25 treatments. Prostate 6000 rad.	RP 11 (4%)	NA	NR
Pilepich [29], 1983, USA, RCT RTOG 75–06: 1976 –1982 RTOG 77–06: 1977–1982	*RTOG 75–06* EBRT prostate and pelvis	131	20 months	66	NR	NR	NR	PLND: Limited 57 (44%) Extended: 7 (5%)	Prostate 6500 rad Pelvis 4000 rad	Neoadjuvant ADT 11.4%	NR	NR
	*RTOG 75-06* EBRT prostate, pelvis & para-aortic	137	21 months	67	NR	NR	NR	Limited 57 (44%) Extended 7 (5%)	Prostate 6500 rad Pelvic LN 4000 rad PA LN: 4000 rad	Neoadjuvant ADT 13.1%	NR	NR
	*RTOG 77-06* EBRT prostate	113	19 months	68	NR	NR	NR	PLND: Limited 59 (52%)	Prostate 6500 rad 180–200 rad/day.	Neoadjuvant ADT 5.3%	NR	NR
	*RTOG 77-06* EBRT prostate and pelvis	106	20 months	66	NR	NR	NR	PLND: Limited 59 (52%) Extended 0	Prostate 6500 rad Pelvic LN 4500–5000 rad 180–200 rad/day.	Neoadjuvant ADT 5.7%	NR	NR

n = number of patients; FU = follow-up; IQR = interquartile range; iPSA = initial Prostate-Specific Antigen; bGS = biopsy Gleason Score; cT = clinical T stage; pN = pathological N stage; RT = radiotherapy; EBRT = external beam radiation therapy; RP = radical prostatectomy; PLND = pelvic lymph node dissection; LN = lymph node; RRP = radical retropubic prostatectomy; NR = not reported; NA = not assessed; RARP = robot-assisted radical prostatectomy; BMI = Body Mass Index; ADT = androgen deprivation therapy.

**Table 2 cancers-14-05667-t002:** Lymphedema rates of included studies.

Study ID	Type of Intervention		N		Prevalence of Lymphedema			*p*-Value
	Intervention	Comparator	Int.	Comp.	Lymphedema Subtype	Intervention	Comparator	
**SURGERY**								
Anscher [12], 1987	RRP ± PLND + adjuvant RT.	RRP ± PLND.	46	113	Not specified	4/46 (9%)	2/113 (2%)	NR
Carlsson [13], 2022	RRP/RARP + PLND	RRP/RARP	437	2578	Lower limb + groin	85/621 (14%)	89/2902 (3%)	<0.001
Chenam [14], 2018	RARP ± limited/extended PLND + pelvic drain.	RARP ± limited/extended PLND + no pelvic drain.	97	92	Lower limb LE	2/97 (2%)	0/92 (0%)	NR
Clark [15], 2003	RRP + e PLND.	RRP + limited PLND.	123 *	123 *	not specified	3/123 (4%), 3/5 occurring on the extended side	2/123 (2%)	NR
Davis [16], 2011	RARP + e PLND.	RARP + limited PLND.	670	261	Lower limb LE	1/670 (0%)	0/261 (0%)	NR
Feicke [17], 2009	RARP + e PLND.	NA	99	NA	Lower limb LE	2/99 (2%)	NA	NA
Kim [18], 2014	RARP + e PLND.	NA	147	NA	Lower limb LE	15/147 (10%),	NA	NA
Mattei [19], 2013	RARP + e PLND.	NA	134	NA	Lower limb LE	1/134 (1%)	NA	NA
Morizane [20], 2018	RARP + e PLND.	RARP + limited PLND.	431	902	not specified	28/431(6%)	7/902 (1%)	*p* < 0.001
Porcaro [21], 2019	RARP + extended PLND.	NA	211	NA	Lower limb LE	5/211 (2%)	NA	NA
					Genital LE	1/211 (0%)	NA	NA
Yuh [22], 2013	RARP + extended PLND.	RARP + limited PLND.	202	204	Lower limb LE	1/202 (0%)	0/204 (0%)	NR
					Genital LE	1/202 (0%)	3/204 (1%)	NR
**RADIATION THERAPY**								
Amdur [23], 1990	EBRT prostate ± pelvis	NA	225	NA	Not specified	2/225 (1%)	NA	NA
Aristizabal [24], 1984	EBRT prostate ± pelvis	NA	218	NA	Lower limb LE	1/218 (0%)	NA	NA
					Genital LE	4/218 (2%)	NA	NA
Borghede [25], 1997	EBRT prostate ± pelvis	NA	184	NA	Lower limb LE	4/184 (2%)	NA	NA
Forman [26], 1985	EBRT prostate + pelvis after staging PLND	EBRT prostate + pelvis without staging PLND	41	199	Genital LE	9/41 (22%)	2/199 (1%)	NA
					Lower limb LE	12/41 (29%)	5/199 (3%)	NA
Perez [27], 1980	EBRT prostate + pelvis after staging PLND	EBRT + pelvic RT without staging PLND	14	181	Lower limb LE	3/14 (21%)	3/181 (2%)	NA
					Genital edema	4/195 (2%)	NR	NA
Pilepich [28], 1981	EBRT prostate + pelvis after staging PLND	EBRT + pelvic RT without staging PLND	31	236	Lower limb LE	8/31(26%)	0/236 (0%)	NA
					Genital edema	6/267 (2%)	NA	NA
Pilepich [29], 1983	RTOG 75-06 PPP Prostate, pelvic and para-aortic irradiation. ± staging PLND	RTOG 75-06 PP Prostate and pelvic irradiation ± staging PLND	137	131	Lower limb LE	6/137 (4%)	11/131 (8%)	*p* = 0.26
					Genital LE	5/137 (4%)	8/131 (6%)	*p* = 0.26
					LE in pts undergoing PLND	Overall, 24/72 (18%)		
	RTOG 77-06 PP Prostate and pelvic irradiation.	RTOG 77-06 P Prostate irradiation	106	113	Lower limb LE	3/106 (3%)	0/113 (0%)	*p* = 0.03
					Genital edema	5/106 (5%)	0/113 (0%)	*p* = 0.03

int. = intervention; comp. = comparator; RRP = radical retropubic prostatectomy; RARP = robot-assisted radical prostatectomy; EBRT = external beam radiation therapy; PLND = pelvic lymph node dissection; RT = radiotherapy; NA = not applicable; NR = not reported; LE = lymphedema. * 123 patients undergoing radical prostatectomy were randomized to an extended node dissection on the right versus the left side of the pelvis with the other side being a limited dissection

## Data Availability

Not applicable.

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
