# Peer review of "The Prevalence of Lower Limb and Genital Lymphedema after Prostate Cancer Treatment: A Systematic Review"

_cancers, 2022, doi:10.3390/cancers14225667_

Round 1

Reviewer 1 Report

Dear Author 

This is a good manuscript about lymphedema survey after radiotherapy or radical prostatectomy for prostate cancer.

Author Response

Thank you for reviewing our manuscript and providing your feedback. 

As suggested, we have performed additional language editing in the revised manuscript. 

Reviewer 2 Report

I had the pleasure to revise this interesting and well-structured review on an undereported complication of prostate cancer treatments.

I found minor spelling errors such as in the abstract (line 34) the word "complication" is lacking and in in the introduction (line 52) a comma instead of a period is needed.

Author Response

Thank your for your careful review. 

We have corrected the spelling errors, according to the reviewer's suggestions. 

Reviewer 3 Report

The authors present an important analysis of the incidence of lymphedema in patients undergoing treatment for prostate cancer. The authors perform a meta-analysis of relevant literature with well-defined inclusion criteria and quantification of biases. The conclusions are generally reasonable and are supported by the studies. However, the following points need to be addressed:

1.      Did the authors also use alternate spellings of lymphedema, such as lymphoedema, that is used more in Europe? Can the authors confirm that such nuances have not biased the analysis towards American patients?

2.      Some acronyms have not been defined before use – RP (line 85), PLND (line 85), RT (line 85), PSA (line 105), GS (line 105), TNM (line 105), TN (line 105). Please check the manuscript thoroughly and define all acronyms (including any I might have missed) before using them. A lot of these seem to be defined in the Results section but they must first be defined in Methods.

3.      How long after intervention were the lymphedema incidence reported? Did all the studies look at a similar timepoint after surgery?

4.      In Table 2, can the authors also include the method by which lymphedema was determined, i.e., operator-defined or patient-defined and also whether objective or subjective criteria were used to define lymphedema?

5.      The authors should include a few sentences acknowledging the generally lower incidence rate of prostate cancer related lymphedema and compare to incidence rates of breast cancer surgery related lymphedema.

6.      The authors should proofread the manuscript for small grammatical and spelling errors.

Author Response

  1. Did the authors also use alternate spellings of lymphedema, such as lymphoedema, that is used more in Europe? Can the authors confirm that such nuances have not biased the analysis towards American patients?

Thank you for this comment. We have indeed tried to keep our literature screening as sensitive as possible to avoid any selection bias: We have used “index terms”, which also include synonyms and alternative spelling) and, more importantly, we have used the term “complication” in our literature search to avoid missing articles that report lymphedema only in the full text results but not in the abstract.

We have included this description in our methodology lines 77-82.

  1. Some acronyms have not been defined before use – RP (line 85), PLND (line 85), RT (line 85), PSA (line 105), GS (line 105), TNM (line 105), TN (line 105). Please check the manuscript thoroughly and define all acronyms (including any I might have missed) before using them. A lot of these seem to be defined in the Results section but they must first be defined in Methods.

Thank you for this valid comments, we have revised the manuscript and defined all acronyms at first use, as suggested by the reviewer.

  1. How long after intervention were the lymphedema incidence reported? Did all the studies look at a similar timepoint after surgery?

This is a very relevant comment. Only one study specially describes the timing of LE, as described in the results section (lines 184). For the other studies, only the duration of follow up has been reported and included in Table 1. In agreement with the reviewer, we have added an extra sentence in the limitations section (lines 317-318), to acknowledge this as a shortcoming.

  1. In Table 2, can the authors also include the method by which lymphedema was determined, i.e., operator-defined or patient-defined and also whether objective or subjective criteria were used to define lymphedema?

Unfortunately, only one study has reported how LE was determined. In the study from Carlsson et., both patient and staff-reported outcomes are provided, as discusses lines 180-187. For all the other studies, it is unknown how LE is determined. Therefore, we cannot include these data in the Table. We have discussed this as the major limitation of the literature (and this review) both in the abstract and in the discussion (line 315-323)

  1. The authors should include a few sentences acknowledging the generally lower incidence rate of prostate cancer related lymphedema and compare to incidence rates of breast cancer surgery related lymphedema.

We have added a sentence about the low incidence lower limb lymphedema compared to upper limb LE in breast cancer (line 297-300). In the next sentences (lines 300-312), we try to elaborate on why these differences may be present.

  1. The authors should proofread the manuscript for small grammatical and spelling errors.

We have reviewed the manuscript for English language, including grammatical and spelling errors.

Reviewer 4 Report

This is a well performed systematic review on the incidence of lymph oedema in patients treated for prostate cancer. 

The review deals with the incidence of lymphoedema. This should be mentioned clearly within the title, abstract and Introduction section of the manuscript. Furthermore, it should be mentioned what are the outcomes (the incidence) of lymph oedema in radical prostatectomy, specifically lymph node dissection, and primary radiotherapy, specifically pelvic node irradiation. This is not separated, but crucial for those who treat patients. This is really mandatory (for urologists and radiotherapists)

So, the paper does not deal with treatment. When as a reader I start reading the manuscript, I hope to find treatment options for lymph oedema, but this is apparently not the subject. Please be very clear in this. 

Author Response

This is a well performed systematic review on the incidence of lymph oedema in patients treated for prostate cancer. 

The review deals with the incidence of lymphoedema. This should be mentioned clearly within the title, abstract and Introduction section of the manuscript. Furthermore, it should be mentioned what are the outcomes (the incidence) of lymph oedema in radical prostatectomy, specifically lymph node dissection, and primary radiotherapy, specifically pelvic node irradiation. This is not separated, but crucial for those who treat patients. This is really mandatory (for urologists and radiotherapists)

So, the paper does not deal with treatment. When as a reader I start reading the manuscript, I hope to find treatment options for lymph oedema, but this is apparently not the subject. Please be very clear in this. 

Thank you for your comments.

As suggested, we have altered our manuscript title, to highlight the focus of this manuscript. We have also adapted the abstract and manuscript to make it clear what the focus of this manuscript is.

Moreover, we now provide the separated prevalence numbers of surgery and radiation therapy more clearly in the abstract (lines 25-30), the results and the discussion (lines 249-253) section.